# Inhibition of Microglial GSK3β Activity Is Common to Different Kinds of Antidepressants: A Proposal for an In Vitro Screen to Detect Novel Antidepressant Principles

**DOI:** 10.3390/biomedicines11030806

**Published:** 2023-03-07

**Authors:** Hans O. Kalkman

**Affiliations:** Child and Adolescent Psychiatry Research Centre, Department of Child and Adolescent Psychiatry and Psychotherapy, Psychiatric University Hospital, University of Zurich, CH-8032 Zurich, Switzerland; hans.kalkman@bluewin.ch

**Keywords:** depression risk factor, microglia, toll-like receptor, GSK3β, NRF2, G_S_-coupled receptor, ketamine, cannabinoid CBR2, psilocybin, 5-HT2B

## Abstract

Depression is a major public health concern. Unfortunately, the present antidepressants often are insufficiently effective, whilst the discovery of more effective antidepressants has been extremely sluggish. The objective of this review was to combine the literature on depression with the pharmacology of antidepressant compounds, in order to formulate a conceivable pathophysiological process, allowing proposals how to accelerate the discovery process. Risk factors for depression initiate an infection-like inflammation in the brain that involves activation microglial Toll-like receptors and glycogen synthase kinase-3β (GSK3β). GSK3β activity alters the balance between two competing transcription factors, the pro-inflammatory/pro-oxidative transcription factor NFκB and the neuroprotective, anti-inflammatory and anti-oxidative transcription factor NRF2. The antidepressant activity of tricyclic antidepressants is assumed to involve activation of G_S_-coupled microglial receptors, raising intracellular cAMP levels and activation of protein kinase A (PKA). PKA and similar kinases inhibit the enzyme activity of GSK3β. Experimental antidepressant principles, including cannabinoid receptor-2 activation, opioid μ receptor agonists, 5HT2 agonists, valproate, ketamine and electrical stimulation of the Vagus nerve, all activate microglial pathways that result in GSK3β-inhibition. An in vitro screen for NRF2-activation in microglial cells with TLR-activated GSK3β activity, might therefore lead to the detection of totally novel antidepressant principles with, hopefully, an improved therapeutic efficacy.

## 1. Introduction

Major depressive disorder is a psychiatric syndrome involving persistent low mood, anhedonia, fatigue, loss of energy, sleep disturbances, and deficits in the cognitive domain, including impaired ability to concentrate, poor attention, and memory [1]. The view on major depression has changed during the last three decades from being a mental disorder caused by a lack of monoamine neurotransmitters [2] to a concept where the mental disorder is secondary to an immune activation in the brain [3,4]. The earliest antidepressants were serendipitously discovered, and, as a common denominator, they all raise the extracellular levels of dopamine, noradrenaline, and serotonin (5-hydroxytryptamine) in the brain [5,6,7]. The indirect monoaminergic activity [8] provides a moderately effective, central anti-inflammatory effect and initiates a slowly-developing improvement in the physical and mental symptoms of depression [9]. Major depression ranks high on the list of diseases posing the strongest burden on society [10], and unfortunately, the global situation has further deteriorated recently due to infections with SARS-Cov2 [11]. The discovery of antidepressants with a novel mode of action has stagnated for many decades, and recent progress is very modest. There is a pressing need to identify novel antidepressant principles and develop alternatives to the currently available insufficiently effective medications. Of course, there has been a been continuous progress in the understanding of the pathophysiology of depression and the mechanism of action of existing antidepressants. In the current review the novel insights in the pathophysiology of major depression and the pharmacology of antidepressants are integrated. Based on this, it is possible to propose a relatively simple in vitro screening method to identify new pharmacological principles with anti-inflammatory and antidepressant potential.

## 2. Activation of Microglia Cells by Risk Factors for Depression

Risk factors for depression (see below) activate the immune system in the brain [4,9,12]. These risk factors activate Toll-like receptors (TLRs) expressed by the brain’s resident immune cells (microglia) and trigger an intracellular signaling pathway involving Nuclear Factor-κB (NFκB) [13] that promotes the transcription of genes that encode pro-oxidative and pro-inflammatory mediators. The serine/threonine kinase glycogen-synthase kinase 3 (GSK3) acts as a master switch between, on the one hand, a pro-inflammatory/pro-oxidative and, on the other hand, a proliferative, protective, and restorative biological program [14,15]. This process, which is discussed in more detail below, is depicted in Figure 1.

The evidence that inflammation plays a subtle role in the pathophysiology of major depressive disorder (MDD) comes from three observations: (1) one-third of those with major depression show elevated peripheral inflammatory biomarkers, even in the absence of an additional medical illness; (2) inflammatory illnesses are associated with greater rates of MDD; and (3) patients treated with pro-inflammatory cytokines are at greater risk for developing major depressive illness (reviewed by [16,17]).

Infection [18] and inflammatory disorders such as IBD (inflammatory bowel disease) [19], metabolic disorders [19,20], cardiovascular disorders [21], neurological disorders [22,23,24,25,26], autoimmune diseases [18], and even alcohol abuse [27,28] are among the likely causes of immune dysfunction that contribute to the pathogenesis of depression [29]. In these diseases, affected cell populations generate and release so-called “danger associated molecular-pattern molecules” (DAMPs), which activate the innate immune system [30]. For instance, DAMPs such as HMGB1, modified lipids, heat shock proteins, S100, or hyaluronan-oligosaccharides activate the same signaling cascade as LPS, a cell wall fragment from Gram-negative bacteria [29,30,31]. Pathogens are recognized by specific immune receptors called Toll-like receptors (TLRs). Like LPS [30], DAMPs activate TLR2 and, in particular, TLR4 [32,33]. Notably, alcohol triggers a TLR4 response too [34].

Inflammation is thought to affect brain signaling causing mood symptoms, cognition dysfunction, and the production of a constellation of symptoms, termed ‘sickness behavior’. Sickness behavior is a set of symptoms (fatigue, immobility, fever, and loss of appetite) induced by an infection and mediated by pro-inflammatory cytokines. It is conceptualized as an adaptive response that enhances recovery by conserving energy to combat acute inflammation [35,36]. Mood symptoms in depression are anxiety, apathy, general discontent, guilt, hopelessness, loss of interest, anhedonia, and sadness [35].

Chronic stress greatly contributes to the development of major depressive disorder [14,37,38,39,40]. The pathophysiological sequelae to chronic stress have been evaluated in great detail in an array of rodent models, including chronic mild stress, social defeat stress, restraint stress, prenatal stress, and others (for review, see [41,42]). In each of these models, microglial cells were activated in a manner that resembled a bacterial infection [41,43,44,45,46,47,48], in particular in mood-relevant areas such as the hippocampus, frontal cortex, nucleus accumbens, and amygdala [31,38,49,50,51]. Chronic stress in laboratory animals thus increased the expression of pro-inflammatory cytokines and chemokines (IL1β, IL6, TNFα, TLR4, CCL2, and CX3CR1) by microglia and elevated the levels of corticosterone and IL6 in the blood circulation [52,53]. In further studies in chronic stress models, it was noted that stress increased inflammation markers (CD14, CD86, and TLR4) on microglia and macrophages [53,54]. Moreover, the morphology of microglia changed from ramified to hypertrophic/activated, with short and thick processes [42,53,55]. Similar to stress, chronic exposure to glucocorticoids also provoked microglial activation in rodents [39]. The process by which stress results in activation of microglia may involve the release of cellular distress signals such as HMGB1 (an agonist at TLR2 and TLR4 [32,56]) and ATP by abnormally active neurons [41,56]. Since attempts to understand the pathophysiology of depression have traditionally focused on neuronal dysfunction, the functionality of other types of brain cells has received (too) little attention [42]. The firm awareness that microglial cells play a crucial role in the development and maintenance of major depression has developed only quite recently [41,42,57,58,59].

Transcription in cells is limited by two important histone acetyltransferases, CBP and p300. In microglia cells, inflammatory signaling is to a large extent determined by the competition for CBP/p300-occupancy between the pro-inflammatory transcription factor NFκB (in particular the p65/RelA subunit) and the anti-inflammatory transcription factors, NRF2 and CREB [60,61,62,63,64]. As mentioned above, the balance between these pro- and anti-inflammatory pathways, which is essential for major depressive disorder [65,66], is influenced by the kinase activity of GSK3. Active GSK3 enhances the affinity of p65 for CBP and p300 [61] and, on the other hand, promotes the nuclear export of NRF2 [67], inhibits the nuclear import of CREB [60], and promotes the catabolism of CREB and NRF2 [68]. Although GSK3 comes in two distinct isoforms, modulation of CREB and NRF2 seems to be mediated by the GSK3β isoform only [60]. NRF2 increases the transcription of genes involved in oxidant-defense [63] and factors that stimulate mitochondrial biosynthesis [66,69]. Downstream of CREB are anti-inflammatory mediators such as brain-derived neurotrophic factor (BDNF), IL1-receptor antagonist (IL1-RA), IL10, and DUSP1 [62,67]. Notably, DUSP1 is an inhibitor of the pro-inflammatory kinase p38 [70,71,72]. Activation of NRF2 provides transcriptional repression of TNFα, IL1β, IL6, IL8, and CCL2 in microglia, monocytes, and macrophages [66,69]. For instance, activation of NRF2 reduced TLR4-induced IL1β, IL6, iNOS, and COX2 expression in primary mouse peritoneal macrophages [66,69]. Stimulation of Toll-like receptors promotes the enzyme activity of GSK3β [14] (for review, see [73]), whereas active GSK3 inhibits the production of the anti-inflammatory cytokine IL10, of IL1-RA, BDNF, and DUSP1 [72,74]. Activators of TLR-signaling are mutant presenilin-1 [75], α-synuclein fibrils [76], ethanol [77], and pathogens [73]. Importantly, repeated stress results in activation of GSK3 [73]. A group of kinases with similar substrate preferences, the so-called ‘AGC’ kinases are able to phosphorylate a serine residue in the N-terminal part of GSK3 (Ser-9 in GSK3β), which generates a pseudo-substrate for GSK3 and thereby inhibits the phosphorylation of other protein substrates (so inhibits GSK3 activity). AGC-kinases include the protein kinases PKA, PKB/Akt, PKC, PKG, S6K, and others [15,78,79]. Data from patients indicate that the cAMP-PKA pathway is hypoactive in MDD [80,81,82], so GSK3 is tendentially hyperactive [83]. This prediction is consistent with observations in MDD patients that both the nuclear translocation of NRF2 and the NRF2-mediated gene transcriptions are diminished [84], whereas biomarkers reflecting oxidative damage are elevated [85,86].

## 3. Classical Antidepressants Activate PKA to Inhibit GSK3β

Interestingly, as in MDD patients, also in the chronic mild stress depression model there is a reduction in cortical cAMP-PKA signaling and consequently a decrease in CREB activation and BDNF transcription [39]. Chronic but not acute treatment with ‘classical’ antidepressants attenuates the decrease in BDNF [8]. Classical antidepressants are typically inhibitors of monoamine reuptake, inhibitors of monoamine metabolism, or antagonists at presynaptic monoamine receptors, so their overall effect is an increase in extracellular levels of monoamine neurotransmitters. It is conceivable that this will lead to stimulation of monoamine receptors on microglia. Some of these receptors (dopamine D1/D5 [76,87,88,89], noradrenaline β2 [90,91], and serotonin 5HT7 [92]) are coupled to G_S_ and generate high cAMP levels. Elevated cAMP levels activate PKA, which in turn inhibits GSK3β. As outlined above and depicted in Figure 1, the balance between pro-inflammatory TLR-NFκB signaling and NRF2/CREB signaling would be shifted away from inflammation (seen as a reduction in pro-inflammatory cytokines) and towards transcription growth factors such as insulin-like growth factor-1, BDNF, and anti-inflammatory cytokines such as IL10 and IL1-RA [93,94]. In accordance with this, it has been proposed that classical antidepressants limit the sequelae of TLR-activation by activation of cAMP-PKA signaling [8,95,96] and reduce the production of pro-inflammatory cytokines and oxygen radicals by microglia cells [52,96,97,98,99,100]. Importantly, chronic antidepressants also ameliorate the behavioral effects of stress [42,52]. So, these antidepressants limit an ongoing inflammation in the brain, and this is presumably the reason why they reduce depression symptoms. However, since the classical antidepressants elevate monoamine levels in the brain, any inflammatory process in the periphery will remain unopposed [67]. It is conceivable that a continuing peripheral inflammation will hinder full recovery and might represent a precipitating factor for relapses [16].

G_S_-coupled monoamine receptors that are expressed by microglial cells are D1 and D5 [88], β1, and β2, and 5HT7 (notably two other G_S_-coupled serotonin receptors, 5HT4 and 5HT6, are not expressed by microglia [101,102]). Direct agonists for these microglial receptors would provide anti-inflammatory and antidepressant activity, possibly with a more rapid onset of action. In this context, it is worth noting that already in 1996, Shimizu and colleagues suggested that activation of 5HT7 receptors might underlie the therapeutic response to classical antidepressants [95]. However, there is no reason for a restriction to monoamine receptors only. Extensive reviews have been published that report on the types of G-protein-coupled receptors (GPCRs) expressed by microglia [101,103,104]. Those GPCRs that couple to G_S_ might offer alternative targets for novel anti-inflammatory and antidepressant drugs. In fact, there is not even a reason to stick to only the cAMP-PKA pathway. Essential is the activity of GSK3β, and this enzyme can be inhibited by, for instance, PKB or PKC. The next paragraphs deal with compound classes with certain evidence for clinical efficacy against depression, for which activation of PKB or PKC could represent the underlying mechanism for therapeutic activity.

## 4. Activation of PKB/Akt to Inhibit GSK3β

Electrical stimulation of the vagus nerve is an exploratory method for the treatment of depression and inflammation [105,106,107,108]. The mode of action involves activation of nicotinic α7 receptors [109,110,111], which results in increases in Ser9-phosphorylation of GSK3β [112,113,114]. This pharmacological response involves a pathway where stimulation of the α7 receptor activates the kinase JAK2, which leads to subsequent activation of PI3K, PKB, and phosphorylation/inhibition of GSK3 [115,116]. Consistent with the inhibition of GSK3, NRF2 signaling was promoted, as seen by increases in gene transcription of antioxidant and anti-apoptotic genes and by a reduction in the levels of pro-inflammatory cytokines [112,115,116]. These effects occur in microglia cells [117,118]. Importantly, treatment with an experimental nicotinic α7 agonist mitigated both the biochemical changes and the depressive behavior of mice subjected to the chronic mild stress procedure [119]. It may be noted that responses to nicotinic-α7 agonists resemble the effects of the anti-inflammatory cytokine IL10. Both activate the JAK2/STAT3 and the PI3K-Akt pathways to improve NRF2-mediated HO-1 (heme oxygenase-1) transcription, and both inhibit NFκB and suppress the hypertrophic/activated microglia polarization [116].

BDNF is produced by neurons [120,121,122], astrocytes [123,124], endothelial cells [93], as well as by reparative microglia cells [125,126]. The receptor for BDNF, TrkB (tropomyosin kinase receptor B), is expressed by neurons [120,122], astrocytes, and microglia [127,128,129,130]. TrkB signaling leads to activation of the PI3K-Akt pathway [122,131,132] and, as a consequence thereof, to inhibition of GSK3β [133] and activation of CREB and NRF2 [134]. Since long, BDNF is suspected to play an important role in the antidepressant effect of both classical and novel antidepressant medications [135,136,137,138,139]. HDAC inhibition by valproate, butyrate, trichostatin, or fingolimod increases the expression of BDNF [121,140,141]. HDAC inhibitors may act as antidepressants [142,143], which is consistent with their proposed effect on the BDNF-TrkB-Akt-GSK3 signal transduction cascade.

## 5. Gq-PLC-PKC-Induced GSK3β-Inhibition

### 5.1. Fatty Acids and Endocannabinoids

Fatty acids and their oxidized metabolites can form endogenous cannabinoids when they are esterified to ethanolamine, glycerin, or amino acids such as glycine or serine. These mediators are produced by astrocytes and microglia [144] and activate several GPCRs [145]. Microglia and macrophages particularly express CBR2 and the ‘orphan’ cannabinoid receptor, GPR18 [144,146,147,148]. Activation of both receptors inhibits the production of pro-inflammatory cytokines and increases the levels of IL10 and IL1-RA [147,148,149]. GPR18 seems to couple to G_S_ [150,151], whereas the CB2 receptor causes PLC activation [147], although G_S_-cAMP-PKA signaling has been reported too [152]. The CB2 receptor is inducible and becomes expressed in microglia cells following inflammation or injury (reviewed by [153]). Overexpression of CBR2 in mice resulted in resistance to chronic mild stress-induced depression [39,154]. The ethanolamines (EA’s) formed from the omega-3 polyunsaturated fatty acids DHA and EPA (DHA-EA and EPEA, respectively) display anti-inflammatory activity in microglia, both in vitro and in vivo [155]. DHA-EA (known also as “synaptamide”) activates cAMP-PKA signaling in microglia and is a potent suppressor of LPS-induced neuroinflammation in mice [156,157,158]. Synaptamide is an agonist at the Gs-coupled orphan receptor GPR110/ADGRF1 [159]. Likewise, the epoxides of EPA and DHA (EEQ and EDP, respectively) can be esterified to ethanolamine, generating the endogenous cannabinoids EEQ-EA and EDP-EA. These products are produced by BV-2 microglial cells and are known to inhibit IL6 and NO production while raising the levels of IL10 [160]. These responses are in part mediated through CBR2 [160], however, any agonistic activity at GPR18 or GPR110 remains to be tested. Such experiments are indicated, as there is an evident discrepancy between the potent anti-inflammatory effects of 19,20-EDP-EA and its weak CB2 affinity in the data reported by McDougle et al. [160]. To summarize, the data collected in this paragraph indicate that agonists for microglial endocannabinoid receptors could represent novel approaches for the treatment of depression.

### 5.2. Opioid μ-Receptor Agonists

Other examples of receptors that are Gq-coupled and activate PLC-PKC are the opioid μ-receptor and the class of 5HT2-receptors (see Table 1).

The endogenous opioid system is known to play a central role in the pathophysiology of affective disorders [175,176,177]. Opioid μ-receptors, in contrast to δ- or κ-opioid receptors, are involved in immune modulation in the brain [178]. Furthermore, depression-scores in healthy subjects were inversely correlated with the binding capacity of an μ-receptor PET-ligand in cortical and subcortical brain regions [179]. Microglia cells express μ-receptors [103,163], and the prototypic opioid receptor agonist morphine decreases the production of IL1β, IL2, TNFα, and IFNγ, and increases TGFβ1 and IL10 [163]. Similarly, tianeptine, a selective μ-receptor receptor agonist [180], inhibited LPS-induced microglia M1-polarization and reduced the LPS-induced expression of IL1β, IL18, IL6, TNFα, and CCL2, as well as the production of NO and ROS [165]. In animal models, tianeptine reversed stress-induced dystrophy of hippocampal dendrites and produced antidepressant-like effects [176]. These effects were blocked by a selective μ-receptor antagonist and were absent in mice with a genetic μ-receptor deficiency [176]. In contrast to morphine, which has off-target pro-inflammatory activity via TLR4 in BV-2 microglia cells [181], tianeptine did not induce tolerance to the analgesic and anti-inflammatory effects [176]. Tianeptine reduces the depression symptoms in patients with mild to moderate-to-severe major depression [167], but given its inherent abuse liability, the risk-benefit ratio of tianeptine remains uncertain [182]. The mixed μ-receptor agonist/κ-antagonist buprenorphine is antidepressant too, and similar to tianeptine, it remains a therapeutic option for patients with treatment-resistant major depression [166,168]. The μ-receptor primarily couples to G_i_/G_0_ but additionally to G_q_-mediated PLC-activation [164]. Activation of PLC leads to PKC activity and thereby, to phosphorylation/inhibition of GSK3.

### 5.3. Hallucinogenic 5HT2 Agonists

Anecdotal reports of the antidepressant activity of psilocybin and LSD [173] have led to the exploration of the antidepressant effects of psilocybin in randomized clinical trials [172,174]. Although the psychedelic effect precludes an effective study blinding, the antidepressant effect was encouraging. The hallucinogenic activity is ascribed to stimulation of central 5HT2A receptors [183,184], and in the neuron-centric view on depression, 5HT2A receptor stimulation is considered the likely mechanism for the improvement of mood [173,185]. Notably, serotonergic hallucinogens are potent activators of the 5HT receptor of the rat stomach fundus [186], a model for 5HT2B receptor activation of PKC [170]. Importantly, the 5HT2B receptor is strongly expressed in microglia [101], and its stimulation suppresses the release of pro-inflammatory cytokines [187]. It is conceivable that the antidepressant activity of psilocybin is related to the activation of 5HT2B receptors on microglia. This would imply that it should be possible to dissociate the hallucinogenic activity from the antidepressant activity by increasing the 5HT2B/5HT2A selectivity.

## 6. Discussion

Stress and infection activate the NFκB pathway in phagocytic cells and lead to the transcription of pro-inflammatory cytokines and oxidizing enzymes such as COX2 and iNOS [44,188,189,190,191]. The opposite process is initiated by the transcription factors CREB and NRF2, which compete with NFκB for access to the gene transcription machinery and activate the production of anti-oxidant proteins and anti-inflammatory cytokines [192,193,194]. Both CREB and NRF2 are stabilized when the pro-inflammatory kinase GSK3β is inhibited. GSK3β becomes blocked when kinases such as PKA, PKB, and PKC phosphorylate a distinct serine residue (Ser-9) in the N-terminal part of GSK3β. Any process that activates PKA, PKB, or PKC therefore has the potential to limit the consequences of stress and infection. In the foregoing sections, several pharmacological principles with antidepressant potential were discussed that, in each case, inhibited GSK3β, either by activating PKA, PKB, or PKC in microglial cells. A screen in microglial cells for compounds and pharmacological principles that inhibit GSK3β or activate CREB and NRF2 transcription has the potential to discover novel antidepressants. The pharmacological principles that were discussed in the foregoing sections were mostly investigated in LPS-stimulated human or rodent microglia cell lines [92,128,195], rat and mouse microglia primary cultures, or primary microglia acutely isolated after in vivo exposure to LPS or stress [94,118]. A good alternative to consider would be to screen induced pluripotent stem cell-derived microglial precursor cells [196]. However, a simple screen in mouse BV-2 microglial cells would already detect α-lipoic acid [197], 9-*cis*-retinoic acid [198], stable cAMP analogues [94,199], PDE4 inhibitors [200,201], agonists of the fatty acid receptor FFAR1/GPR40 [202], the ‘specialized pro-resolving mediators’ lipoxin A4, resolvin D1 and resolvin E1 [59,203,204,205,206], and inhibitors of the enzyme soluble epoxide-hydrolase [207]. Presumably, even ketamine might be detected as a hit.

### 6.1. Ketamine

The mechanism by which ketamine exerts its antidepressant effects is still a matter of debate [208,209]. The two mechanisms that are discussed are glutamate NMDA-receptor blockade and opioid μ-receptor activation. The issue with NMDA-blockade is that other NMDA blockers do not induce a rapid antidepressant effect (for instance, memantine [210]), whereas the ketamine-metabolite 2R,6R-HNK provoked rapid antidepressant-like activity in preclinical models that was independent of NMDA-blockade [211]. Moreover, ketamine was still ‘antidepressant’ in transgenic mice that lack the essential NR1-subunit of the NMDA channel, whereas glycine-site agonists that *activate* the functioning of the NMDA-channel are also antidepressants (reviewed in [209]). Ketamine has a significant affinity for opioid receptors and is antinociceptive [210]. In a pilot study in humans with major depression, the antidepressant but not the dissociative effects of ketamine were blocked by naltrexone [212]. Notably, the antinociceptive effects of 2R,6R-HNK were *not* affected by the opioid μ-receptor antagonist naltrexone [213]. Independent of the precise mechanism of action, ketamine induced a robust inhibitory phosphorylation of GSK3 in preclinical models, whereas the antidepressant-like effect of ketamine was lost in transgenic mice that carried a non-inhibitable GSK3-variant [214,215]. Consistent with this activity, ketamine increased the expression of BDNF [216,217], increased the expression of the NRF2-target gene heme oxygenase-1 [218], suppressed chronic stress-induced depressive behaviors and pro-inflammatory cytokine levels [219,220,221], as well as LPS-induced depressive behaviors [222]. In a human monocyte cell line, ketamine inhibited NFκB and reduced pro-inflammatory cytokine levels [223,224]. Ketamine administration to humans before or during surgery inhibited the increase in post-operative IL6 and TNFα levels [225,226]. Similarly, in two small studies in patients suffering from MDD, blood IL6 levels decreased after ketamine treatment in the responders but not in the non-responders [227,228]. These results suggest that suppression of inflammation is relevant for the antidepressant effects of ketamine. The remaining question is: by what mechanism does ketamine provoke inhibition of GSK3β. An interesting proposal was made by Wray and co-authors [124]. G_S_ is singly palmitoylated on its N-terminus, which targets the protein to lipid rafts. Ketamine and 2R,6R-HNK are lipophilic and remove G_s_ from the lipid raft. This has a positive effect on the interaction between G_S_ and adenylate cyclase and thus on the production of cAMP, the activation of PKA, and the CREB-target gene BDNF [82,124]. The authors consider it likely that other lipid-soluble anesthetics act similarly to ketamine and 2R,6R-HNK. This proposal is consistent with the observation that intravenous and inhaled anesthetics (ketamine, propofol, opioids, isoflurane, sevoflurane, desflurane, etc.) to varying extents upregulate heme oxygenase-1 expression [218]. The ketamine-induced displacement of G_S_ from the lipid rafts occurs much more rapidly than with, for instance, tricyclic antidepressants [82] and could be an explanation for ketamine’s faster onset of antidepressant activity. It is conceivable that microglial cells are the targets for these effects [41,229,230].

### 6.2. GSK3β Blockade by Antipsychotics

Although dopamine D2 receptor antagonists (antipsychotics) are known to elevate the levels of Ser-9 phosphorylated GSK3β in the brain [231], such compounds are not genuinely known to act as antidepressants. A conceivable explanation for this apparent discrepancy is provided hereafter.

Dopamine and serotonin not only influence microglia cells, they also modulate the function of neurons. However, the downstream signaling pathways in neurons are distinctly different from those in microglia. Activation of dopamine D2 receptors expressed on neurons (by, for instance, amphetamine) leads to activation of GSK3. The signaling cascade involves the formation of a molecular complex consisting of β-arrestin2 (βArr2), PKB, and the phosphatase PP2A [232]. Formation of this complex leads to PP2A-mediated deactivation of PKB and thus to the disinhibition of GSK3 [232]. Conversely, D2-receptor antagonists such as haloperidol, raclopride, and atypical antipsychotics such as clozapine, risperidone, olanzapine, quetiapine, and ziprasidone (their common profile is dual antagonism of 5HT2A receptors and D2-receptors [233,234]) all induce inhibition of GSK3, owing to the inhibitory phosphorylation of GSK3 by PKB [235,236]. Similar to dopamine, activation of neuronal 5HT2A receptors involves the formation of a molecular complex, here consisting of βArr2, PKB, and Src [237] and ultimately results in activation of GSK3 [238]. In agreement, 5HT2A receptor antagonists provoke an inhibition of GSK3 [239]. Therefore, a GSK3 inhibitor will block amphetamine-induced mania (GSK3 inhibition in neurons) and will generate antidepressant responses (via GSK3 inhibition in microglia). The solution to the apparent discrepancy mentioned above is quite simple. Whereas a GKS3 inhibitor will presumably limit both the manic and depressive symptoms, a D2-receptor antagonist will affect the signaling in neurons [231], but will not produce a profound antidepressant effect (see Figure 2). Under the assumption that this explanation is proven to be correct, it would emphasize the point that only GSK3 blockade in microglia results in antidepressant activity.

### 6.3. Conclusions

Depression is a major public health concern. The risk-benefit ratio of the currently available antidepressants is quite unsatisfactory, since either their therapeutic effectiveness is insufficient or they cause unacceptable side effects. Consequently, there is a pressing need to discover better treatments. Literature data indicate that risk factors for depression initiate an infection-like inflammation in the brain that involves activation of microglial Toll-like receptors and glycogen synthase kinase-3β (GSK3β). GSK3β activity alters the balance between two major competing transcription factors, the pro-inflammatory/pro-oxidative transcription factor NFκB and the neuroprotective, anti-inflammatory, and anti-oxidative transcription factor NRF2. The negative consequences of depression risk factors could thus be counteracted by procedures that lead to GSK3β inhibition. As an example, the antidepressant activity of tricyclic antidepressants is assumed to involve activation of G_S_-coupled microglial receptors, raising intracellular cAMP levels, and activation of protein kinase A (PKA). PKA inhibits the enzyme activity of GSK3β. Importantly, PKA is just one of a series of AGC-kinases that is able to inhibit GSK3β. Literature information is provided to show that each pharmacological principle for which evidence for antidepressant activity in humans exists results in the inhibition of GSK3β in microglial cells, albeit via different signaling cascades and involving different AGC-kinases. Ketamine and lipid-soluble anesthetics act somewhat distinct in that they do lead to GSK3β inhibition; however, not via a pharmacological effect but in response to a physical property (lipid solubility) that alters the lipid raft composition. Furthermore, it is argued that in order to achieve antidepressant activity, GSK3β inhibition has to occur in microglia rather than in neurons. Based on these considerations, an in vitro screen testing for GSK3β inhibition/NRF2 activation in microglial cells with TLR4-stimulated GSK3β activity might ultimately lead to the detection of novel antidepressant principles. This would, hopefully, end the stagnation in the discovery and development of safe and effective treatments for major depressive disorders.

## Figures and Tables

**Figure 1 biomedicines-11-00806-f001:**
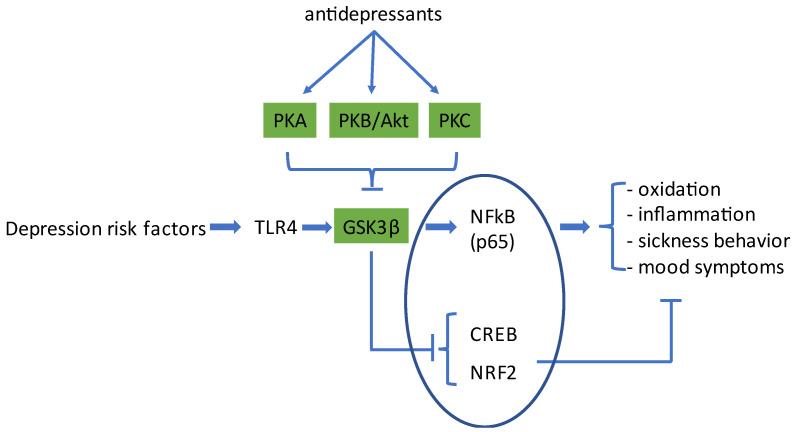
Schematic representation of the process by which risk factors for depression increase inflammatory and pro-oxidative processes in the brain and lead to physical and psychological symptoms of depressive disorder. Depression risk factors activate Toll-like receptor-4 (TLR4) on microglia cells, which causes activation of the kinase GSK3β and an increase in gene transcription by NFκB (p65/RelA). Antidepressant compounds, via activation of different members of the AGC protein-kinase family, including protein kinase A (PKA), PKB/Akt, and PKC, inhibit the enzyme activity of GSK3β. This, on the one hand, inhibits NFκB-signaling and, on the other hand, increases gene transcription by CREB and NRF2, two transcription factors that promote the transcription of neuroprotective, anti-inflammatory, and anti-oxidant proteins. This way, antidepressants neutralize the negative effects evoked by TLR4 activation. CREB: cyclic-AMP responsive element binding protein; NRF2: nuclear factor erythroid 2-related factor 2. The transcription factors are enclosed in an ellipse. For further details, the reader is referred to the text.

**Figure 2 biomedicines-11-00806-f002:**
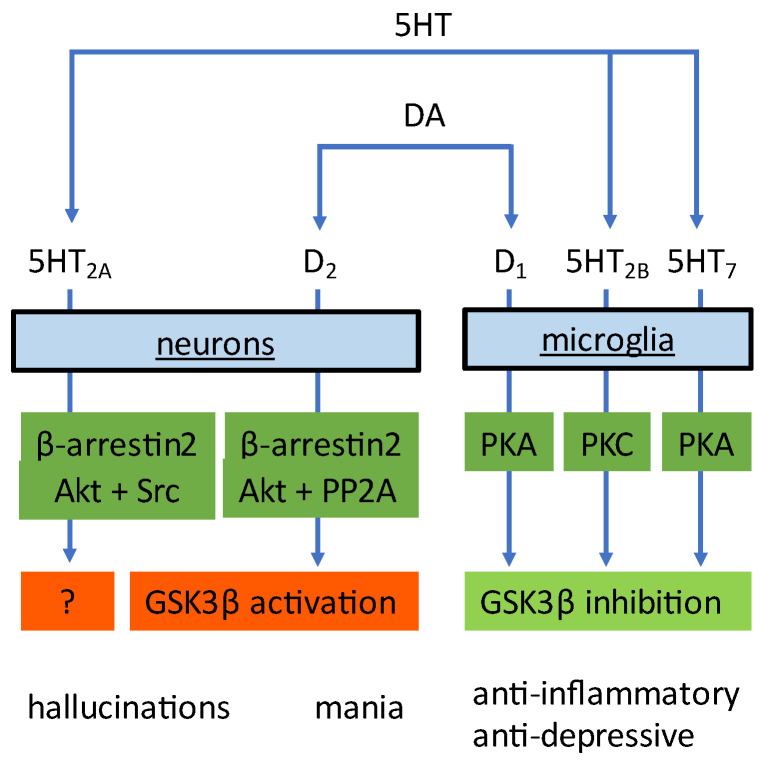
Dopamine (DA) and serotonin (5HT) activate distinct receptor subtypes and signaling pathways in microglia and neurons. In microglial cells, the functional outcome is an inhibition of GSK3β, whereas activation of the D2-receptor in neurons results in activation of GSK3β. Whether activation of the 5HT2A receptor in neurons leads to GSK3β activation has not been investigated.

**Table 1 biomedicines-11-00806-t001:** Gq-coupled receptors that mediate anti-inflammatory and antidepressant activity.

Receptor	Citations for Expression by Microglia	Citations for Gq-PLC-PKC Coupling	Citations for Inflammatory Cytokines Decrease	Citation for Antidepressant Activity
Cannabinoid CBR2	[146,153]	[147,161]	[148,149,161]	[144,162]
Opioid μ-receptor	[103,163]	[164]	[165]	[166,167,168]
Serotonin 5HT2B	[101,102]	[169,170,171]	[101,102]	[172,173,174]

Literature citations for the expression of the cannabinoid receptor CBR2, the opioid μ receptor, and the serotonin receptor 5HT2B by microglia cells. Agonists at these receptors activate phospholipase-C (PLC) and protein kinase-C (PKC), which results in inhibition of brain inflammation and improvement of depression symptoms.

## Data Availability

Not applicable.

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
