# Peer review of "Inhibition of Microglial GSK3β Activity Is Common to Different Kinds of Antidepressants: A Proposal for an In Vitro Screen to Detect Novel Antidepressant Principles"

_biomedicines, 2023, doi:10.3390/biomedicines11030806_

Round 1

Reviewer 1 Report

Kalkman in the present review article entitled ‘An in vitro screen for detection of novel antidepressants and antidepressant principles’, explored how the pathophysiology of depressive disorder, in particular how activation of microglia cells, has been proposed as a relatively simple in vitro screening method to identify new pharmacological principles with an anti-inflammatory and antidepressant potential. Results showed that risk factors for major depression activate Toll-like receptors on microglia cells, which induces physical and psychological symptoms of depression, therefore addressing that GSK3β inhibitory activity in TLR-stimulated microglial cells/cell lines may have antidepressant effects.

The main strength of this paper is that it addresses an interesting and timely question, providing an overview of possible anti-inflammatory / antidepressant pharmacological principles that could be discovered in a simple in vitro screen of TLR-activated microglia. In general, I think the idea of this manuscript is really interesting and the authors’ fascinating observations on this timely topic may be of interest to the readers of Biomedicines. However, some comments, as well as some crucial evidence that should be included to support the author’s argumentation, needed to be addressed to improve the quality of the manuscript, its adequacy, and its readability prior to the publication in the present form.

Please consider the following comments:

A graphical abstract that will visually summarize the main findings of the manuscript is highly recommended.

Abstract: According to the Journal’s guidelines, the abstract should be a total of about 200 words maximum; please correct the actual one. Also, in my opinion, Author should consider rephrasing this section. According to the Journal’s guidelines, the Abstract should contain most of the following kinds of information in brief form. Please, consider giving a more synthetic overview of the paper's key points: I would suggest rephrasing the results and conclusion to make them clear for readers to understand.

I would ask the Author to clarify the criteria used for studies’ collection in this review: he should specify the requirements used to decide whether a study met the inclusion/exclusion criteria of the review, describe whether they included a balanced coverage of all information that is actually available, whether he has included the most recent and relevant studies and enough material to show the development and limitations in this field of interest. Finally, I believe that he should briefly present results of all statistical syntheses conducted.

The objectives of this study are generally clear and to the point; however, I believe that there are some ambiguous points that require clarification or refining. In my opinion, author should be explicit regarding how the studies selected for this review have assessed molecular and biological mechanisms that are involved in pathomechanisms of depressive disorder.

Introduction: The ‘Introduction’ section is well-written and nicely presented, with a good balance of descriptive text and information about pathophysiology of major depressive disorder. Nevertheless, I believe that a more detailed overview about neurobiological pathomechanisms and core features of this disorder will provide a better and more accurate background, because as it stands, this information is not highlighted in the text. I suggest to begin with a theoretical explanation of depressive disorder and the role of specific brain areas in regulating neurotransmitters release, like prefrontal cortex,. In this regard, I would suggest to add more information on pathological neural substrates of depression disorder, for example focusing on ‘Dissecting Neurological and Neuropsychiatric Diseases: Neurodegeneration and Neuroprotection’ and on structural as well as functional abnormalities of prefrontal cortex that may affect patients’ cognitive impairments (https://doi.org/10.3390/biomedicines10123189). In my opinion, authors could further explore relationship between the molecular regulation of higher-order neural circuits and neuropathological alterations in this neuropsychiatric disorder (https://doi.org/10.3390/biomedicines10081897), in order to provide more insights on dysregulation of neurotransmitters in depressive disorder.

In my opinion, I think the ‘Conclusions’ paragraph would benefit from some thoughtful as well as in-depth considerations by the Author, because as it stands, it is very descriptive but not enough theoretical as a discussion should be. Author should make an effort, trying to explain the theoretical implication as well as the translational application of his research.

I would ask the Authors to include a proper and defined ‘Limitations and future directions’ section before the end of the manuscript, in which he can describe in detail and report all the technical issues that could be brought to the surface.

Finally, what is the take-away message from this review article? It ends rather abruptly with no summary, no suggested directions or immediate challenges to overcome, no call to action, no indications of things we should stop trying, and only brief mention of alternative perspectives. What do the authors want us to take away from this paper?

References: Author should consider revising the bibliography, as there are several incorrect citations. Indeed, according to the Journal’s guidelines, they should provide the abbreviated journal name in italics, the year of publication in bold, the volume number in italics for all the references.

I hope that, after these careful revisions, the manuscript can meet the Journal’s high standards for publication. I am available for a new round of revision of this article. 

Best regards,

Reviewer

Author Response

I would like to thank the referee for taking the time to read my manuscripts and for his suggestions where to improve it!

Graphical abstract: In principle, Figure 1 is the graphical abstract.

Abstract size: I have shortened and rewritten the abstract.

Criteria for citations: I have not employed a formal search strategy, and it was not my intention to be comprehensive on every detail. The articles that were selected for citation, were 1) those that support the statement in the manuscript, 2) those that I have found and read, and 3) that seemed scientifically sound. In an area as huge as the MDD literature, I have undoubtedly missed some important papers, but that should not limit the overall validity of the message of the manuscript, namely that antidepressants lead to inhibition of GSK3 in microglia, and that TLR-activated microglia can be used to screen for novel antidepressant mechanisms.

Study objectives: This is in my opinion the same point as above. The patho-mechanism of MDD is presumably an infection-like activation of the brain innate immune system (microglia) with consequences on neuronal function, circuit function and behavioural output.

More details on MDD in the Introduction (Symptoms, Brain Regions): I have added a summary of the clinical symptoms of MDD (although I consider this rather superfluous: it is a repetition of something that is in every MDD-review paper, and in Wikipedia and DSM). I am hesitant to include the two citations from the Bologna EEG-group. One is an article on EEG markers in patients with brain injury and coma. The other paper is a review on EEG alterations in mental health disorders. In the latter case, the results for MDD are descriptive and correlative, and may help in the diagnosis. But, I do not quite see, how these impact on the message of my manuscript, which basically is describing cellular pathology and pharmacological mechanisms. Furthermore, it is outside my competence to write with authority about detailed roles of specific brain areas (I am a preclinical pharmacologist, not an MD).    

Conclusions: I have rewritten the conclusion section.

To add limitations and future directions is difficult. I present a hypothesis that fits a huge amount of literature, and as future direction I propose a screen in microglia cells.

The take home message (that is hopefully recognised by pharmaceutical companies) is that novel antidepressant principles can be detected by a relatively simple screen in cultured microglia cells. This might end the stagnation in the discovery of truly novel antidepressant treatments.

Citations: I have used the MDPI style that can be downloaded from the internet. Concerning incorrect citations, I have read each and every publication that is cited, and to the best of my knowledge, I have always correctly quoted their data and conclusions.

Reviewer 2 Report

In this manuscript, the author proposed a central role of inflammation in the actions of antidepressants. The hypothesis is generally clearly described and well supported by the evidence that the author listed. I believe the manuscript will make an important contribution to the field. Nevertheless, I have several concerns for the author to address, which may help enhance the manuscript.

1), the 1st sentence of introduction is inaccurate: the inflammation hypothesis is but one hypothesis proposed to explain the actions of antidepressants, there are also other hypotheses, such as the neurogenesis hypothesis, the glucocorticoid hypothesis, the microbiota hypothesis, and so on, this sentence should be revised.

2), lines 42-43: "discovery of antidepressants...has stagnated for many decades, and recent progress is very modest" may also be inaccurate, SSRI, SNRI, and NaSSa were primarily discovered in the past 2-3 decades or so.

3), lines 75-76, the term "sickness behavior" should be explained in more details. The same comment applies to "mood symptoms" in figure 1.

4), the description of the risk factors show in Figure 1 should be in more details, right now, it is somewhat superficial.

5), can the authors, for instance, add a box and provide some general explanation the key terms mentioned in Figures 1 and 2? more comprehensive information on PKA, PKB, PKC, TLRs, etc, will make the manuscript readable to a large audience.

6), IL-6 is considered an anti-inflammatory cytokine in many situations.

7), lines 95-97: "microglial cells play a decisive role" may be an overstatement.

8), the author mentioned "classical antidepressants" and "novel antidepressants", they should clarify what they meant, depending on the argument that they make.

 9), section 6.1, for the actions of ketamine, what is the time scale of the inflammation effect as well as the effect on microglial cells? can these effects explain the fast actions of ketamine? please clarify.

Author Response

Like for referee 1, I would like to thank the referee 2 for his time and efforts to read and comment my manuscript.

Point 1) Of course, I am aware of the other hypotheses. Neurogenesis becomes impaired during an inflammation, so it is impaired during chronic depression too. The hypothesis is no longer a favourite, because compounds that reduce neurogenesis do not always cause depression, and improved antidepressants does not necessarily improve depression. The microbiota hypothesis is in a way a variation of the infection/inflammation hypothesis (pathogenic gut microbes provoke inflammation in the GI-tract and may proceed to the blood and further). The glucocorticoid hypothesis led to the development of CRF-antagonists. These compounds failed in depression (at least up to now). I am hesitant to include and discuss alternative hypotheses, because it detracts from the main message of the manuscript.   

Point 2) The first SSRI (zimelidine) was around when I moved to Switzerland (1984) and was shortly thereafter followed by fluoxetine. SNRIs are in principle cleaned-up tricyclics, not a new discovery. NaSSa is a marketing terminology for mianserin and mirtazapine. I have used mianserin for my PhD thesis (>40 years ago), and mirtazapine was developed for patent reasons to replace mianserin. The only really new antidepressant in the last years is agomelatine. This compound is however only moderately effective, and suffers from hepato-toxicity (the low benefit/risk ratio is the reason why it has never been submitted for review to the FDA). So, there truly is a stagnation in innovation. 

Point 3) (Sickness behaviour) I have added these details to the introduction paragraph.

Point 4) I have included a section on risk factors in the Introduction, and refer to the text in the figure legend.

Point 5) It is not clear what kind of information the referee is looking for. These are kinases that ‘put’ a phosphate group on Serines or Threonines of proteins. This is in most cases activating, but in case of GSK3 it is inhibitory. I can add in the legend that these are Ser/Thr kinases with a similar substrate specificity. Concerning TLR, I think this becomes clearer with a small section in the Introduction. But in my opinion, this is common knowledge.

Point 6) This is true. The response depends on the receptor-combination that is activated. IL6 in combination with the hydrolysed circulating IL6-receptor-trunk causes “trans-activation” and this is highly pro-inflammatory. The IL6-receptor becomes hydrolised during inflammation, so that the beneficial growth-stimulating effect of IL6 gets lost. It would be better if during clinical research, the circulating hydrolised IL6-receptor moiety would be measured, but currently almost every paper reports the IL6-level, only. An IL6 antibody is therapeutic because it prevents the formation of the IL6+hydrolised IL6-receptor combination. So, in the context of depression, IL6 levels predict a pro-inflammatory effect.  

Point 7) 'Decisive' changed to 'crucial'

Point 8) I now avoid the term classical (replaced by monoamine modifying). With “novel” I mean truly new mechanistic principles. 

Point 9) The effect of ketamine in redistributing Gs from the lipid rafts into the cell membrane, is much faster than occurs for e.g. tricyclics. This is discussed by Schappi and Rasenick (citation is in the reference list). A side-by-side comparison for the timing of anti-inflammatory effects has not been performed (would be a relevant study to do!).

Round 2

Reviewer 1 Report

The author did an excellent job clarifying all the questions I have raised in my previous round of review. Currently, this paper entitled ‘Inhibition of microglial GSK3β activity is common to different kinds of antidepressants. Proposal for an in vitro screen to detect novel antidepressant principles’, is a well-written, timely piece of research that improves the understanding of how activation of microglia cells could be used as a relatively simple in vitro screening method to identify new pharmacological principles with an anti-inflammatory and antidepressant potential.

Overall, this is a timely and needed work. It is well-researched and nicely written, so I believe that this paper does not need a further revision, therefore the manuscript meets the Journal’s high standards for publication.

I am always available for other reviews of such interesting and important articles.

Thank You for your work,

Reviewer

Reviewer 2 Report

thank the author for addressing my concerns